# Physics-informed Dynamics Representation Learning for Parametric PDEs

## Abstract

While physics-informed neural networks have achieved remarkable progress in modeling dynamical systems governed by partial differential equations (PDEs), their ability to generalize across different scenarios remains restricted. To address this limitation, we present PIDO, a novel physics-informed neural PDE solver that demonstrates robust generalization across various aspects of PDE configurations, including initial conditions, PDE coefficients, and training time horizons. PIDO leverages the shared intrinsic structure inherent to dynamical systems with varying properties by projecting the PDE solutions into a latent space via auto-decoding and subsequently learning the dynamics of these latent embeddings conditioned on the PDE coefficients. However, the inherent optimization challenges associated with physics-informed loss present substantial obstacles to integrating such latent dynamics models. To tackle this issue, we adopt a novel perspective by diagnosing these challenges within the latent space. This approach enables us to enhance both temporal extrapolation ability and training stability of PIDO via simple yet effective regularization techniques, ultimately leading to superior generalization performance compared to its data-driven counterpart. The effectiveness of PIDO is validated on diverse benchmarks, including 1D combined equations and 2D Navier-Stokes equations. Moreover, we investigate the transferability of its learned representations to downstream tasks like long-term integration and inverse problems.

## 1 Introduction

Partial differential equations (PDEs) constitute the cornerstone of comprehending complex systems and forecasting their behavior. Recent years have witnessed a surge in the effectiveness of deep learning methods for solving PDEs (Yu et al., 2018; Kovachki et al., 2021; Brandstetter et al., 2021). Among these, Physics-Informed Neural Networks (PINNs) have emerged as a burgeoning paradigm (Raissi et al., 2019). PINNs leverage Implicit Neural Representations (INRs) to parameterize PDE solutions, enabling them to effectively bridge data with mathematical models and tackle high-dimensional problems. This unique characteristic has led to their widespread adoption in a variety of applications , including computational fluid dynamics (Raissi et al., 2020), photonic structures design (Ma et al., 2021) and material science simulations (Zhang et al., 2022).

A key advantage of PINNs lies in their ability to be trained by enforcing PDE-based constraints even in the absence of exact solutions. This flexibility proves valuable in real-world settings where perfect data might not be available. However, this very benefit comes at a cost. Each instance of PINNs is trained tailored to a specific configuration of initial and boundary conditions, PDE coefficients, geometries, and forcing terms. Modifying any of these elements necessitates retraining, resulting in significant computational inefficiency. In addressing this obstacle, Neural Operators (NOs) have been proposed as a promising solution (Li et al., 2020b; Lu et al., 2021a). NOs aim to tackle the so-called *parametric* PDEs by learning to map variable condition entities to corresponding PDE solutions.

Despite their success, several limitations hinder the generalization ability of NOs. First, some NO architectures restrict the choice of grids. This limitation leads to challenges when encountering input (Lu et al., 2021a; Wang et al., 2021b) or output (Li et al., 2020b) grids that differ from those used in training. Secondly, for time-dependent PDEs, NOs are typically trained to provide predictions within a fixed time horizon, limiting their applicability in scenarios requiring broader temporal coverage. Finally, while NOs may generalize well to specific types of variable conditions, their ability to handle

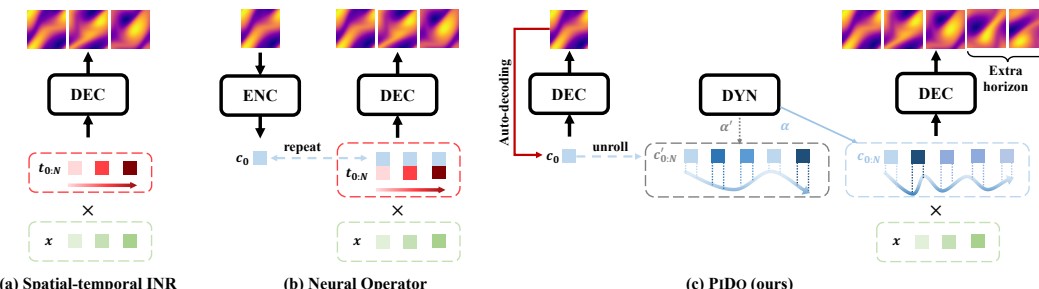

Figure 1: Physics-informed neural PDE solvers. Colorful squares denote spatial/temporal coordinates and embeddings of PDE solutions at different times, while curves represent continuous trajectories of embeddings unrolled by the dynamics model, with colors indicating different values. The symbol "$\times$" denotes Cartesian product. PIDO handles multiple types of conditions, including initial conditions, coefficients $\alpha$ and training time horizons, by learning evolution of embeddings conditioned on $\alpha$.

concurrent variations in multiple types of conditions has not been thoroughly validated. This is particularly concerning in domains like coefficient-aware dynamics modeling, where models must simultaneously adapt to varying initial conditions and PDE coefficients (Brandstetter et al., 2021).

This paper tackles the limited generalization ability in physics-informed neural PDE solvers. To this end, we introduce the **P**hysics-**I**nformed **D**ynamics representati**O**n learner (PIDO), a versatile framework capable of generalizing across different types of variables in PDE configurations, including initial conditions, PDE coefficients and training time horizons. The key to PIDO's success lies in its two core components as shown in Figure 1. First, the grid-independent *spatial representation learner* exploits the intrinsic structure shared between dynamical systems governed by different PDE coefficients. This is achieved by learning a mapping between the solution space and a low-dimensional latent space via auto-decoding (Park et al., 2019). This latent space captures the essential representations of the solutions, enabling generalization to unseen initial conditions. Inspired by Explicit Dynamics Modeling (EDM) (Yin et al., 2022; Wan et al., 2022), the *temporal dynamics model* then learns the coefficient-aware evolution of latent embeddings using Neural ODEs (Chen et al., 2018), which are known for their exceptional ability to extrapolate beyond the training horizon.

While previous EDM methods are typically trained in a data-driven manner, their performance relies heavily on dataset size. In contrast, we focus on a physics-informed setting, where PIDO is optimized to satisfy governing PDEs without relying on real data. However, the physics-informed loss is known to face optimization challenges (Krishnapriyan et al., 2021; Wang et al., 2021a), leading to two key issues in its integration with EDM: instability during training and degradation in time extrapolation. To address these challenges, we adopt a novel perspective by diagnosing and tackling them within the latent space. By projecting high-dimensional data into low-dimensional representations, we identify two problematic latent behaviors: overly complex dynamics and latent embedding drift. Based on these insights, we propose two regularization techniques-*Latent Dynamics Smoothing* and *Latent Dynamics Alignment*-to improve training stability and extrapolation ability, respectively. Overall, these strategies enhance PIDO's generalization ability compared to its data-driven counterparts. Our contributions can be summarized as follows:

- We propose a novel physics-informed learning framework that achieves robust generalization across diverse variables in PDE configurations.

- We diagnose the learning difficulties of physics-informed dynamics models within the latent space and address them using latent dynamics smoothing and latent dynamics alignment, resulting in improved generalization performance compared to the data-driven counterpart.

- We demonstrate the effectiveness of PIDO on extensive benchmarks including 1D combined equation and 2D Navier-Stokes equations, and explore the transferability of PIDO's learned representations to downstream tasks including long-term integration and inverse problems.

## 2 RELATED WORK

**Spatial-temporal Implicit Neural Representations** are powerful paradigm to model continuous signals in 3D shape representation learning and PDEs solving (Sitzmann et al., 2020; Fathony et al.,

Table 1: Comparisons of Physics-informed Neural PDE Solvers.

| Method | Generalize to unseen initial conditions | Generalize to unseen PDE coefficients | Time extrapolation | Data-free | Flexibility on grid choice |
|---|---|---|---|---|---|
| PINN (Raissi et al., 2019) | ✗ | ✗ | ✗ | ✓ | ✓ |
| PI-DeepONet (Wang et al., 2021b) | ✓ | ✓ | ✗ | ✓ | ✗ |
| MAD (Huang et al., 2022) | ✓ | ✓ | ✗ | ✓ | ✓ |
| PINODE (Sholokhov et al., 2023) | ✓ | ✗ | ✓ | ✗ | ✗ |
| DINo (YIN ET AL., 2022) | ✓ | ✗ | ✓ | ✗ | ✓ |
| PiDo (Ours) | ✓ | ✓ | ✓ | ✓ | ✓ |

2020). They train a neural network to map spatial-temporal coordinates to PDE solutions as

$$\mathcal{G} : (t, \boldsymbol{x}) \to \boldsymbol{u}(t, \boldsymbol{x}), \tag{1}$$

enabling the model to query any point without being constrained by the resolution of a fixed grid. Taking advantage of this differentiable parameterization, physics-informed neural networks (PINNs) embed PDEs as soft constraints to guide the learning process (Yu et al., 2018; Raissi et al., 2019). This framework is appealing in many realistic situations as it can be trained in the absence of exact PDE solutions. However, it is well known that the PDE-based constraints suffer from ill-conditioned training dynamics (Wang et al., 2021a). Many recent works have devoted efforts to alleviate the optimization difficulty and to improve training efficiency with loss weight balancing (Wang et al., 2022b;a; Yao et al., 2023), curriculum learning (Krishnapriyan et al., 2021) and dimension decomposition (Cho et al., 2023). Another limitation of PINNs lies in its generalization ability, as the model can be only applied to a predefined set of PDE coefficients and initial/boundary conditions.

**Neural Operators** attempt to learn a mapping between two infinite-dimensional function space as

$$\mathcal{G} : (\mathcal{A} \times \mathbb{R}) \to \mathcal{U}, \quad (\boldsymbol{a}, t) \to \mathcal{G}(\boldsymbol{a}, t) = \boldsymbol{u}(t). \tag{2}$$

In the context of parametric PDEs, $\boldsymbol{a} \in \mathcal{A}$ is a function characterizing initial conditions or PDE coefficients and $\boldsymbol{u} \in \mathcal{U}$ denotes the corresponding solution. As per the inherent design, NOs are grid-independent (Li et al., 2020a;b; Lu et al., 2021a). However, many NOs do not have full flexibility on the spatial discretization. For example, DeepONets (Lu et al., 2021a) use an INR to represent the continuous solution, but reply on a fixed discretization of the input function $\boldsymbol{a}$. Additionally, they require a large number of parameter-solution pairs to learn the solution operator. To address the need of solution data, PI-DeepONet (Wang et al., 2021b) proposes to incorporate neural operators with physics-informed training. It can be seen as an extension to PINNs by conditioning the output linearly on the embeddings of input parameters encoded by a branch network. However, this linear aggregation strategy limits its capability in handling complex problems (Lanthaler et al., 2022). As an alternative approach, Meta-Auto-Decoder (MAD) learns an embedding for each input parameter via auto-decoding (Huang et al., 2022). Another line of work (Li et al., 2021) approximates the PDE-based loss with finite difference method based on the outputs of NOs, which is prone to the discretization error. Finally, a key limitation of NOs for time-dependent PDEs is their inherent design for prediction within a specific horizon, $[0, T_{tr}]$, which restricts generalization beyond the time $T_{tr}$.

**Explicit Dynamics Modeling** learns the derivative of solutions with respective to time instead of directly fitting solution values at different time steps as spatial-temporal INRs and NOs do. The solution at a given time can be obtained with numerical integration, which can be formalized as

$$\mathcal{G} : \boldsymbol{u} \to \frac{\partial \boldsymbol{u}}{\partial t}, \quad \boldsymbol{u}(t) = \boldsymbol{u}(0) + \int_{\tau=0}^{t} \mathcal{G}(\boldsymbol{u}(\tau)) d\tau. \tag{3}$$

One kind of EDM is autoregressive models (Greenfeld et al., 2019; Brandstetter et al., 2021). They learn to predict the solution at $t + \delta t$ based on the current solution at $t$ with $\mathcal{G} : \boldsymbol{u}(t) \to \boldsymbol{u}(t + \delta t)$, which equals to approximating the time derivative with finite difference $\frac{\boldsymbol{u}(t+\delta t) - \boldsymbol{u}(t)}{\delta t}$. Another kind of EDM learns the time derivative with Neural ODEs (Chen et al., 2018). Neural ODEs can be queried at any time step and have been widely applied in continuous-time modeling (Quaglino et al., 2019; Ayed et al., 2019). EDM has shown superior extrapolation ability beyond training time interval compared with INRs and NOs in solving initial value problems (Yin et al., 2022; Wan et al., 2022; Serrano et al., 2023). In spite of its efficacy, EDM has two main limitations. First, most EDM methods are agnostic to the underlying PDEs, restricting them to modeling a single type of dynamics. Second, the training of EDM can be data-intensive, especially when modeling multiple dynamics. In this case, the required dataset size grows exponentially with the number of PDE coefficients.

Our proposed method builds upon EDM and addresses these challenges by incorporating physics-informed training. This approach leverages the strengths of both methods (cf. Table 1). Furthermore, we show that the latent space introduced by EDM offers a novel perspective for diagnosing learning difficulties in physics-informed loss. While prior work has explored combining physics-informed training with EDM, these efforts have limitations. PINODE (Sholokhov et al., 2023), for example, utilizes an auto-encoder structure confined to a fixed grid. Additionally, its physics-informed training requires the input data to be sampled from a pre-defined analytical distribution that is similar to the true PDE solutions distribution (see Appendix B for details). This requirement is often impractical in real-world scenarios. In contrast, our method does not require any prior assumptions about the data distribution. Another work (Wen et al., 2023) cannot be trained solely with the physics-informed loss.

## 3 METHOD

n this section, we first outline the problem setting in Section 3.1, followed by the architectural design of PIDO in Section 3.2. Next, we introduce the physics-informed training adapted for EDM in Section 3.3 and analyze the related learning challenges within the latent space in Section 3.4.

### 3.1 PROBLEM SETUP

We focus on the time-dependent parametric PDEs which can be formulated as

$$
\begin{aligned}
\frac{\partial \boldsymbol{u}(t, \boldsymbol{x})}{\partial t} + \mathcal{L}^{\boldsymbol{\alpha}}(\boldsymbol{u}(t, \boldsymbol{x})) &= 0, & (t, \boldsymbol{x}) &\in [0, T] \times \Omega, \\
\mathcal{B}(\boldsymbol{u}(t, \boldsymbol{x})) &= 0, & (t, \boldsymbol{x}) &\in [0, T] \times \partial\Omega, \\
\boldsymbol{u}(t = 0, \boldsymbol{x}) &= \boldsymbol{\phi}(\boldsymbol{x}), & \boldsymbol{x} &\in \Omega,
\end{aligned}
\tag{4}
$$

where $\mathcal{L}^{\boldsymbol{\alpha}}$ is the differential operator parameterized by coefficients $\boldsymbol{\alpha}$ and $\boldsymbol{u} : [0, T] \times \Omega \rightarrow \mathbb{R}^n$ is the solution. Let $\boldsymbol{u}_t \triangleq \boldsymbol{u}(\boldsymbol{t})$ denotes the state of interest at each time step $t$. Our goal is to solve the parametric PDEs by learning the map $\mathcal{G}^* : (t, \boldsymbol{\phi}, \boldsymbol{\alpha}) \rightarrow \boldsymbol{u}_t$ from parameters to solutions. The model is trained on a set of parameters $\{(\boldsymbol{\phi}_i, \boldsymbol{\alpha}_i)\}$ sampled from distribution $\Phi$. At test time, we evaluate the model on unseen initial conditions and coefficients sampled from the same distribution. In addition, we assess the model's temporal extrapolation capability. To achieve this, the training horizon is confined to the interval $[0, T_{tr}]$, where $T_{tr} < T$, and the corresponding test horizon, denoted as $[0, T_{ts}]$, spans the entire time interval with $T_{ts} = T$.

### 3.2 MODEL ARCHITECTURE

The proposed PIDO tackles the dynamics modeling task defined in Equation (4) by approximating the solution map $\mathcal{G}^* : (t, \boldsymbol{\phi}, \boldsymbol{\alpha}) \rightarrow \boldsymbol{u}_t$ with

$$
\tilde{\boldsymbol{u}}_t(\boldsymbol{x}) = \mathcal{D}(\boldsymbol{c}_t, \boldsymbol{x}), \text{ where } \boldsymbol{c}_t = \mathcal{E}(\boldsymbol{\phi}) + \int_{\tau=0}^{t} \mathcal{F}(\boldsymbol{c}_\tau, \boldsymbol{\alpha}) d\tau.
\tag{5}
$$

PIDO achieves this with two key components as shown in Figure 1. First, the spatial representation learner establishes a mapping between the data space and a low-dimensional representation space. It consists of an encoder $\mathcal{E}$, which transforms initial conditions $\boldsymbol{\phi}$ into the latent space via auto-decoding (Park et al., 2019), and a decoder $\mathcal{D}$, which represents continuous solutions given a latent embedding $\boldsymbol{c}_t$ at time step $t$. Second, the temporal dynamics model $\mathcal{F}$ learns the coefficients-aware evolution of latent embeddings starting from the initial embedding $\mathcal{E}(\boldsymbol{\phi})$.

**Spatial representation learner** aims to capture the essential representations of PDE solutions in a low-dimensional latent space. It achieves this by learning a decoder $\mathcal{D}$ which can accurately reconstruct solution trajectories $\boldsymbol{u}$ from low-dimensional embeddings $\boldsymbol{c}$. We parameterize the decoder $\mathcal{D}$ with an INR, which approximates the solution $\boldsymbol{u}$ conditioned on both spatial coordinates and embeddings, resulting in $\tilde{\boldsymbol{u}}(\boldsymbol{x}) = \mathcal{D}(\boldsymbol{c}, \boldsymbol{x})$. Benefiting from this parameterization, PIDO achieves grid-independence (see Section 2) and can readily compute the spatial derivatives of $\tilde{\boldsymbol{u}}$ through Automatic differentiation (AD) (Baydin et al., 2018), which are crucial for physics-informed training.

Given a learned decoder $\mathcal{D}$, the encoder $\mathcal{E}$ is defined via auto-decoding. Specifically, the encoder $\mathcal{E}$ identifies the corresponding embedding $\boldsymbol{c}$ of spatial observation $\boldsymbol{u}$ (or initial condition $\boldsymbol{\phi}$) through an

optimization process. This process minimizes the reconstruction error between $\boldsymbol{u}$ and $\tilde{\boldsymbol{u}} = \mathcal{D}(\boldsymbol{c})$ by updating a learnable $\boldsymbol{c}$. In essence, the auto-decoding seeks the optimal embedding $\boldsymbol{c}^*$ that captures the essential information within $\boldsymbol{u}$ necessary for the decoder to accurately reproduce the original observation. Mathematically, this encoder $\mathcal{E}$ can be formulated as

$$\boldsymbol{c}^* = \mathcal{E}(\boldsymbol{u}), \text{ where } \boldsymbol{c}^* = \underset{\boldsymbol{c}}{\arg\min}\, \mathbb{E}_{\boldsymbol{x} \in \Omega} \|\boldsymbol{u}(\boldsymbol{x}) - \mathcal{D}(\boldsymbol{c}, \boldsymbol{x})\|^2. \tag{6}$$

Here the expectation (denoted by $\mathbb{E}$) is taken over spatial coordinates $\boldsymbol{x}$ sampled the domain $\Omega$. In practice, this optimization process can be efficiently achieved by updating the latent embedding $\boldsymbol{c}$ (typically initialized with zeros in our experiments) with a few steps of gradient descent.

Our method distinguishes itself from prior works (Wang et al., 2021b; Huang et al., 2022) by learning the latent space of entire trajectories, not just initial conditions. This enables us to thoroughly capture the intrinsic structure of solution space, leading to enhanced generalization across initial conditions.

**Temporal dynamics model** leverages a Neural ODE to learn the evolution of latent embeddings as

$$\frac{\partial \boldsymbol{c}_t}{\partial t} = \mathcal{F}(\boldsymbol{c}_t, \boldsymbol{\alpha}), \quad \boldsymbol{c}_0 = \mathcal{E}(\boldsymbol{\phi}), \tag{7}$$

where $\mathcal{F}$ is a neural network that predicts the time derivative of $\boldsymbol{c}$. The initial embedding $\boldsymbol{c}_0$ is obtained by encoding the initial condition $\boldsymbol{\phi}$ through $\mathcal{E}$. This continuous formulation of latent dynamics allows our model to compute the embeddings at arbitrary time steps through numerical integration and to extrapolate beyond the training horizons. Our approach departs from previous works (Yin et al., 2022; Wan et al., 2022) by conditioning the prediction of $\mathcal{F}$ on $\boldsymbol{\alpha}$. This empowers the model to effectively capture the rich variety of dynamics governed by different PDE coefficients.

### 3.3 MODEL TRAINING

Most existing EDM methods are predominantly data-driven, relying on extensive datasets of precise solutions for training. However, in many applications, generating sufficient data through repeated simulations or experiments is prohibitively expensive. This problem becomes especially acute in systems with multiple parameters, as the amount of data required scales exponentially with the number of parameters. Such constraints motivate the exploration of physics-informed training, which directly incorporates physical laws into the training process, bypassing the need for large datasets.

**Physics-informed loss for EDM.** Given a pair of sampled parameters $\{(\boldsymbol{\phi}^i, \boldsymbol{\alpha}^i)\}$, we first obtain the initial embedding $\boldsymbol{c}_0^i = \mathcal{E}(\boldsymbol{\phi}_i)$ with auto-decoding as defined in Equation (6). We then optimize $\mathcal{D}$ to reduce the reconstruction error of initial conditions in the auto-decoding process through

$$l_{\text{IC}}(\boldsymbol{\theta}_D, \boldsymbol{\phi}^i) \triangleq \mathbb{E}_{\boldsymbol{x} \in \Omega} \|\boldsymbol{\phi}^i(\boldsymbol{x}) - \mathcal{D}(\boldsymbol{c}_0^i, \boldsymbol{x})\|_2^2, \text{ where } \boldsymbol{c}_0^i = \underset{\boldsymbol{c}}{\arg\min}\, \mathbb{E}_{\boldsymbol{x} \in \Omega} \|\boldsymbol{\phi}^i(\boldsymbol{x}) - \mathcal{D}(\boldsymbol{c}, \boldsymbol{x})\|^2. \tag{8}$$

While the above loss function involves a nested minimization problem, we employ an efficient approximation in practice. Instead of solving for the optimal embedding $\boldsymbol{c}_0^i$ exactly, we update the embedding from its previous value using a single gradient descent. This simplifies Equation (8) as:

$$l_{\text{IC}}(\boldsymbol{\theta}_D, \boldsymbol{c}_0^i, \boldsymbol{\phi}^i) \triangleq \mathbb{E}_{\boldsymbol{x} \in \Omega} \|\boldsymbol{\phi}^i(\boldsymbol{x}) - \mathcal{D}(\boldsymbol{c}_0^i)(\boldsymbol{x})\|_2^2, \tag{9}$$

Then, we leverage the dynamics model $\mathcal{F}$ to unroll the trajectories from initial embedding $\boldsymbol{c}_0^i$. This provide us with the embedding $\boldsymbol{c}_t^i$ at a sampled time step $t \in [0, T_{tr}]$ and its corresponding prediction $\boldsymbol{u}_t^i(\boldsymbol{x}) = \mathcal{D}(\boldsymbol{c}_t^i, \boldsymbol{x})$. Since the exact solution is unavailable, we optimize the PDE residuals instead:

$$\frac{\partial \boldsymbol{u}_t^i(\boldsymbol{x})}{\partial t} + \mathcal{L}^{\boldsymbol{\alpha}^i}(\boldsymbol{u}_t^i(\boldsymbol{x})) = \frac{\partial \boldsymbol{u}_t^i(\boldsymbol{x})}{\partial \boldsymbol{c}_t^i} \frac{\partial \boldsymbol{c}_t^i(\boldsymbol{x})}{\partial t} + \mathcal{L}^{\boldsymbol{\alpha}^i}(\boldsymbol{u}_t^i(\boldsymbol{x})) = \frac{\partial \boldsymbol{u}_t^i(\boldsymbol{x})}{\partial \boldsymbol{c}_t^i} \mathcal{F}(\boldsymbol{c}_t, \boldsymbol{\alpha}^i) + \mathcal{L}^{\boldsymbol{\alpha}^i}(\boldsymbol{u}_t^i(\boldsymbol{x})). \tag{10}$$

Here, the time derivative of $\boldsymbol{c}_t^i(\boldsymbol{x})$ is approximated by $\mathcal{F}$ (with parameters $\boldsymbol{\theta}_{\mathcal{F}}$) conditioned on PDE coefficients $\boldsymbol{\alpha}^i$. Note that $\boldsymbol{c}_t^i$ is not a trainable embedding. Instead, it's obtained through integration from $\boldsymbol{c}_0^i$ as in Equation (5). Taking altogether, we derive the PDE residual loss as

$$l_{\text{PDE}}(\boldsymbol{\theta}_{\mathcal{F}}, \boldsymbol{\theta}_D, \boldsymbol{c}_0^i, \boldsymbol{\alpha}^i) \triangleq \mathbb{E}_{\boldsymbol{x} \in \Omega, t \in [0, T_{tr}]} \|\frac{\partial \boldsymbol{u}_t^i(\boldsymbol{x})}{\partial \boldsymbol{c}_t^i} \mathcal{F}(\boldsymbol{c}_t, \boldsymbol{\alpha}^i) + \mathcal{L}^{\boldsymbol{\alpha}^i}(\boldsymbol{u}_t^i(\boldsymbol{x}))\|_2^2. \tag{11}$$

As we parameterize the decoder $\mathcal{D}$ with an INR, the term $\partial \boldsymbol{u}_t^i(\boldsymbol{x})/\partial \boldsymbol{c}_t^i$ and spatial derivatives of $\boldsymbol{u}_t^i(\boldsymbol{x})$ involved in $\mathcal{L}^{\boldsymbol{\alpha}^i}(\boldsymbol{u}_t^i(\boldsymbol{x}))$ can be calculated with AD.

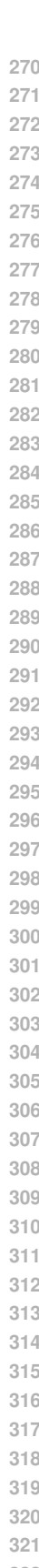

Figure 2: Training instability arises from overly complex dynamics. We randomly sample 3 dimensions from the 128-dim embeddings. We mark training steps in green and test steps in red.

Similarly, we enforce the boundary conditions with the following loss

$$l_{\text{BC}}(\boldsymbol{\theta}_{\mathcal{F}}, \boldsymbol{\theta}_D, \boldsymbol{c}_0^i) \triangleq \mathbb{E}_{\boldsymbol{x} \in \Omega, t \in [0, T_{tr}]} \| \mathcal{B}(\boldsymbol{u}_t^i(\boldsymbol{x})) \|_2^2. \quad (12)$$

Finally, the overall physics-informed objective function can be formalized as

$$l_{\text{PI}}(\boldsymbol{\theta}_{\mathcal{F}}, \boldsymbol{\theta}_D, \boldsymbol{c}_0) \triangleq \mathbb{E}_{(\boldsymbol{\phi}^i, \boldsymbol{\alpha}^i) \sim \Phi} [l_{\text{IC}}(\boldsymbol{\theta}_D, \boldsymbol{c}_0^i, \boldsymbol{\phi}^i) + l_{\text{BC}}(\boldsymbol{\theta}_{\mathcal{F}}, \boldsymbol{\theta}_D, \boldsymbol{c}_0^i) + l_{\text{PDE}}(\boldsymbol{\theta}_{\mathcal{F}}, \boldsymbol{\theta}_D, \boldsymbol{c}_0^i, \boldsymbol{\alpha}^i)]. \quad (13)$$

### 3.4 Diagnosing Physics-informed Optimization in the Latent Space

While physics-informed training eliminates the need for exact PDE solutions, it suffers from optimization difficulties (Krishnapriyan et al., 2021; Wang et al., 2021a), presenting two key challenges in integration with EDM: instability during training and degradation in time extrapolation. Nevertheless, the latent space introduced by EDM offers a novel perspective for diagnosing and addressing these issues. By capturing essential information in low-dimensional representations, it facilitates a more straightforward analysis. Notably, this framework allows us to identify latent behaviors responsible for each challenge and mitigate them using simple but effective regularization in latent space.

#### 3.4.1 Stabilizing the Training Process through Latent Dynamics Smoothing

In contrast to data-driven methods, physics-informed loss imposes stricter constraints on the time step size to ensure stable training. If the time step is too large, training often collapses into trivial solutions as shown in Figure 2(a), where the loss is minimized on training steps but remains high on unseen time points close to the initial conditions. Consequently, despite small training losses, the predictions deviate from the ground truth because information fails to propagate from the initial conditions to subsequent steps (Wang et al., 2022a). To resolve this, a smaller time step size is often required, which, in turn, significantly increase computational costs and exacerbate the training difficulty.

As an alternative, we explore this issue within the latent space. Our findings, as illustrated in Figure 2(a), reveal that the model tends to learn excessively fluctuating latent dynamics from all possible dynamics that minimize training loss. This leads to a complex loss distribution along the time axis, which exhibits poor generalization to unseen points. As a result, the step size constraint becomes even stricter for our method compared to other physics-informed approaches. We attribute this to the increased flexibility introduced by the latent dynamics model, which is trained without direct supervision in latent space. While data-driven EDM methods utilize a similar framework, they do not encounter this issue, as they leverage embeddings from pretrained encoder-decoder networks as ground truth for training the dynamics model (Yin et al., 2022). In contrast, our method provides the dynamics model with only indirect supervision through the PDE loss in data space.

To alleviate this issue, we introduce the *Latent Dynamics Smoothing* regularization, which guides the model to favor simpler dynamics. Inspired by Finlay et al. (2020), this regularization mitigates rapid local changing in the predicted trajectories by constraining the time derivative of dynamics model, given by $\frac{\partial \mathcal{F}(\boldsymbol{c}_t, \boldsymbol{\alpha})}{\partial t} = \nabla \mathcal{F}(\boldsymbol{c}_t, \boldsymbol{\alpha}) \cdot \frac{\partial \boldsymbol{c}_t}{\partial t} = \nabla \mathcal{F}(\boldsymbol{c}_t, \boldsymbol{\alpha}) \cdot \mathcal{F}(\boldsymbol{c}_t, \boldsymbol{\alpha})$, where $\nabla \mathcal{F}(\boldsymbol{c}_t, \boldsymbol{\alpha})$ is the

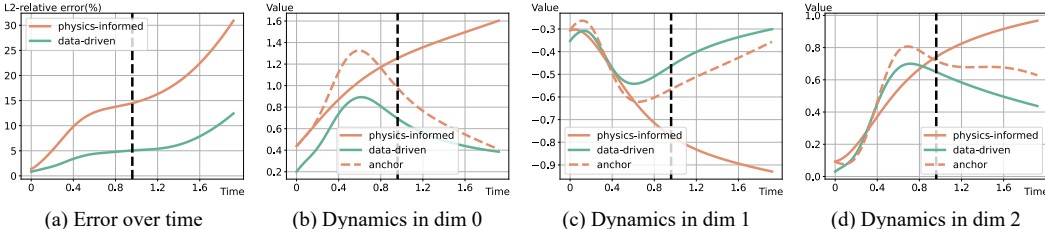

| (a) Error over time | (b) Dynamics in dim 0 | (c) Dynamics in dim 1 | (d) Dynamics in dim 2 |

Figure 3: Time extrapolation degradation arises from latent embedding drift. We take the **CE3** setting as an example. The black dash line separates training and testing horizon. For visualization, we randomly sample 3 dimensions from the 64-dim embeddings.

Jacobian matrix of $\mathcal{F}$ with respective to $\boldsymbol{c}_t$. Specifically, we apply the latent dynamics smoothing regularization $R_S(\boldsymbol{\theta}_{\mathcal{F}})$ to the training time points, where

$$R_S(\boldsymbol{\theta}_{\mathcal{F}}) = \|\mathcal{F}(\boldsymbol{c}_t, \boldsymbol{\alpha})\|_2^2 + \|\nabla \mathcal{F}(\boldsymbol{c}_t, \boldsymbol{\alpha})\|_F^2 = \|\mathcal{F}(\boldsymbol{c}_t, \boldsymbol{\alpha})\|_2^2 + \mathbb{E}_{\boldsymbol{\epsilon} \sim \mathcal{N}(0,1)} \|\boldsymbol{\epsilon}^T \nabla \mathcal{F}(\boldsymbol{c}_t, \boldsymbol{\alpha}) \boldsymbol{\epsilon}\|_2^2. \quad (14)$$

As demonstrated in Figure 2(b), this regularization effectively prevents overly complex dynamics, resulting in a smoother temporal loss distribution without necessitating a reduction in time step size.

### 3.4.2 IMPROVING TIME EXTRAPOLATION VIA LATENT DYNAMICS ALIGNMENT

The next challenge we investigate is the degradation of the model's time extrapolation ability when trained with physics-informed loss compared to data-driven approaches (Figure 3(a)). By analyzing the evolution of latent embeddings over extended time horizons, we trace this issue to the phenomenon of latent embedding drift, where the embeddings progressively move outside their typical range as time advances, as depicted in Figure 3. This drift becomes particularly problematic when extrapolating beyond the training horizon, causing the embeddings to deviate from the training distribution, ultimately leading to poor performance at later time steps.

We attribute the latent embedding drift to the inconsistency between the supervision signals of the initial embeddings $\boldsymbol{c}_0$ and the later ones $\boldsymbol{c}_t$. To be specific, the initial embeddings are guided by both the exact initial conditions and the PDE loss, whereas the later embeddings rely solely on the PDE loss for supervision. To bridge this gap, we propose to utilize predicted solutions $\tilde{\boldsymbol{u}}_t = \mathcal{D}(\boldsymbol{c}_t)$ as pseudo labels in the absence of exact PDE solutions. While these pseudo labels do not provide additional information in the data space, the embeddings obtained through encoding them, defined as $\tilde{\boldsymbol{c}}_t = \mathcal{E}(\tilde{\boldsymbol{u}}_t) = \mathcal{E}(\mathcal{D}(\boldsymbol{c}_t))$, does not exhibit the drift problem as shown in Figure 3. Thus they can serve as effective anchors to regularize $\boldsymbol{c}_t$ (Figure 4). Note that although $\tilde{\boldsymbol{c}}_t$ and $\boldsymbol{c}_t$ represent the same states $\tilde{\boldsymbol{u}}_t$, they might not be identical because the neural network is not inherently an one-to-one mapping. The differences arise from their respective training processes. Specifically, $\boldsymbol{c}_t$ is unrolled from $\boldsymbol{c}_0$ to satisfy the PDE loss, potentially leading to a distribution shift from that of $\boldsymbol{c}_0$. In contrast, $\tilde{\boldsymbol{c}}_t$ are obtained via auto-decoding from the data space (similar to $\boldsymbol{c}_0$), resulting in a distribution closer to the initial embeddings. Therefore, we can mitigate the drift problem by aligning latent embeddings $\boldsymbol{c}_t$ with anchor embeddings $\tilde{\boldsymbol{c}}_t$ via the *Latent Dynamics Alignment* regularization

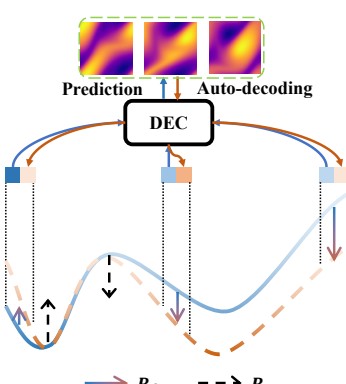

Figure 4: Regularization. Blue curve: dynamics learned with PDE loss; Orange curve: dynamics obtained by auto-decoding the predicted solutions.

$R_A(\boldsymbol{\theta}_{\mathcal{F}}) = \|\boldsymbol{c}_t - \tilde{\boldsymbol{c}}_t\|_2$. Note that this alignment is achieved with minimal impact on the predicted solution, as both embeddings correspond to the same output.

## 4 EXPERIMENTS

We begin by introducing the benchmarks in Section 4.1. Section 4.2 presents the main results of our study. Finally, Section 4.3 explores the transferability of the pre-trained PIDO to downstream tasks.

Table 2: Results on the test set of 1D and 2D benchmarks. We report the $L_2$ relative error (%) over the training horizon (IN-T) and the subsequent duration (OUT-T). The best results are **bold-faced**.

| | CE1 | | CE2 | | CE3 | | NS1 | | NS2 | |
|---|---|---|---|---|---|---|---|---|---|---|
| MODEL | IN-T | OUT-T | IN-T | OUT-T | IN-T | OUT-T | IN-T | OUT-T | IN-T | OUT-T |
| PI-DEEPONET | 4.18 | 8.61 | 17.17 | 36.16 | 7.57 | 15.74 | 23.57 | 36.10 | 29.86 | 47.10 |
| PINODE | 10.44 | 24.75 | 11.03 | 28.69 | 18.21 | 39.41 | 16.44 | 53.52 | 17.56 | 46.67 |
| MAD | 3.98 | 9.32 | 12.00 | 27.97 | 6.78 | 17.10 | 14.85 | 30.50 | 16.95 | 33.49 |
| PIDO | **1.48** | **2.24** | **3.02** | **7.15** | **3.19** | **8.08** | **2.35** | **5.43** | **4.59** | **10.02** |

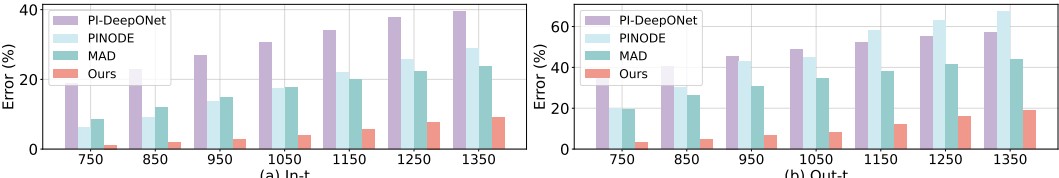

Figure 5: **NS2** test performance vs. Reynolds numbers.

## 4.1 BENCHMARKS

**1D combined equations.** We consider the family of PDEs:

$$\frac{\partial u}{\partial t} + 2u\frac{\partial u}{\partial x} - \alpha_0\frac{\partial^2 u}{\partial x^2} + \alpha_1\frac{\partial^3 u}{\partial x^3} = 0, \quad u(t = 0, x) = \phi(x), \tag{15}$$

where $x \in [0, L_x]$ and $t \in [0, T]$. This equation encompasses several fundamental physical phenomena, namely nonlinear advection, viscosity, and dispersion. It is a combination of Burgers' equation (when $\alpha_1 = 0$) and Korteweg–De Vries (KdV) equation (when $\alpha_0 = 0$). We aim to predict $u(t, x)$ given varying $\phi(x)$ and $\boldsymbol{\alpha} = (\alpha_0, \alpha_1)$. We consider three scenarios: • **CE1** for Burgers' equation with $\boldsymbol{\alpha} = (0.1, 0)$; • **CE2** for KdV equation with $\boldsymbol{\alpha} = (0, 0.05)$; and • **CE3** for combined equation with $\boldsymbol{\alpha} \in \{(\alpha_0, \alpha_1) | 0 < \alpha_0 \le 0.4, 0 < \alpha_1 \le 0.65\}$. More details can be found in Appendix C.1.

**2D Navier-Stokes equations.** We consider the 2D Navier-Stokes equations, which describe the dynamics of a viscous and incompressible fluid. The equations are given by

$$\frac{\partial w}{\partial t} + \boldsymbol{u} \cdot \nabla w - \frac{1}{\alpha}\Delta w - f = 0,$$
$$w = \nabla \times \boldsymbol{u}, \quad \nabla \cdot \boldsymbol{u} = 0, \quad w(t = 0, \boldsymbol{x}) = \phi(\boldsymbol{x}), \tag{16}$$

where $\boldsymbol{x} \in [0, 1]^2$, $w$ is vorticity, $\boldsymbol{u}$ is the velocity field and f is the forcing function. We consider the long temporal transient flow with the forcing term $f(\boldsymbol{x}) = 0.1(\sin(2\pi(x_1 + x_2)) + \cos(2\pi(x_1 + x_2)))$ following previous works (Li et al., 2020b; Yin et al., 2022). The Reynolds number, denoted as $\alpha$, serves as an indicator of the fluid viscosity. The modeling of fluid dynamics becomes increasingly challenging with higher Reynolds numbers, since the flow patterns transition to complex and chaotic regimes. We investigate the prediction of vorticity dynamics under varying initial conditions $\phi$ and Reynolds numbers $\alpha$ with two scenarios: • **NS1** focuses on a fixed $\alpha = 1000$, while • **NS2** considers varying Reynolds numbers with $\alpha \in [700, 1400]$. See Appendix C.1 for more details.

**Tasks and metrics.** For each problem, we create training and test sets by randomly sampling $N_{tr}$ and $N_{ts}$ parameter-solution pairs $\{(\phi_i, \boldsymbol{\alpha}_i, \boldsymbol{u}_i)\}$, respectively. The exact solution $\boldsymbol{u}_i$ is only used for evaluation. For each prediction $\hat{\boldsymbol{u}}_i$, we assess the $L_2$ relative error, expressed as $\|\hat{\boldsymbol{u}}_i - \boldsymbol{u}_i\|_2 / \|\boldsymbol{u}_i\|_2$, over the full time interval $[0, T]$, subdivided into the training horizon $[0, T_{tr})$ (denoted as *In-t*) and the subsequent duration $[T_{tr}, T]$ (denoted as *Out-t*).

## 4.2 MAIN RESULTS

We compare PIDO with PI-DeepONet (Wang et al., 2021b), PINODE (Sholokhov et al., 2023) and MAD (Huang et al., 2022) on the benchmarks detailed in Section 4.1 (see implementation details in Appendix C.2 and computational efficiency in Appendix C.3). For PINODE, which requires input data sampled from the exact solution distribution (unavailable here), we use the distribution of initial conditions as a substitute because these are the only data accessible (see Appendix B for details).

Performance of all methods on the test set is reported in Table 2. Additional results (training set performance, sample efficiency analysis and visualizations) are available in Appendix A.

**Generalizing across initial conditions.** Focusing on scenarios with fixed coefficients, our method achieves the lowest error in test set *In-t*. Specifically, it surpasses the second best method by significant margins: 63% for **CE1**, 72% for **CE2** and 84% for **NS1**. These results highlight PIDO's effectiveness in handling diverse initial conditions within these benchmarks. Notably, the efficacy of PIDO in modeling complex dynamical systems becomes more pronounced when applied to 2D problems.

**Generalizing across PDE coefficients.** We evaluate the models' ability to handle unseen PDE coefficients in scenarios like **CE3** and **NS2**. For PI-DeepONet and PINODE, we provide coefficients $\alpha$ as additional inputs alongside the initial conditions. This allows them to make predictions conditioned on both factors. In this setting, our PIDO demonstrates remarkable robustness to changes in PDE coefficients. Figure 5 presents the relative error for various test Reynolds numbers in **NS2** scenario. We observe that as the Reynolds numbers increase, PIDO consistently maintains a low solution error. See Appendix A.2 for PIDO's extrapolation ability beyond the training distribution.

**Generalizing beyond training horizon.** To assess the models' capability to predict past the training horizon, we adopt the auto-regressive evaluation strategy from Wang & Perdikaris (2023) for PI-DeepONet and MAD. This approach iteratively extends forecasts by using the final state predicted in the training interval as the initial condition for the next one. Across all scenarios, PIDO demonstrates clear superiority over the baseline methods on *Out-t*. While PINODE utilizes a Neural ODE capable of integrating beyond the training horizon, its performance suffers in practice. We hypothesize this limitation stems from its input data distribution failing to capture the true distribution of the solutions.

**Comparison with the data-driven approach.** We compare PIDO with the data-driven counterpart, DINo (Yin et al., 2022), in Table 3. Despite achieving a slightly higher training error than DINo trained with the full dataset, PIDO outperforms it on the test set, demonstrating a superior ability to adapt to unseen initial conditions. We attribute this improvement to the incorporating with physics-informed training, which ensures our model produces physically plausible predictions and reduces overfitting to noise and anomalies in the data. Furthermore, DINo's performance significantly degrades with smaller training sets, emphasizing the limitations of data-driven approaches in achieving robust generalization with limited data. See Appendix A.3 for Comparisons with more data-driven baselines.

Table 3: Comparison with DINo trained with different sub-sampling ratios of training set. $L_2$ relative error in **NS1** is reported (%).

| MODEL | DATASET | TRAIN | | TEST | |
|---|---|---|---|---|---|
| | | IN-T | OUT-T | IN-T | OUT-T |
| PIDO | - | 1.81 | 4.10 | **2.35** | **5.43** |
| DINo | 100% | **1.42** | **3.32** | 4.26 | 5.73 |
| DINo | 50% | 1.14 | 4.90 | 5.25 | 8.76 |
| DINo | 25% | 0.82 | 7.52 | 7.37 | 13.58 |
| DINo | 12.5% | 0.47 | 13.59 | 15.12 | 31.74 |

**Ablations on regularization methods.** We provide ablation studies to verify the effectiveness of two regularization methods. We present the results in the test set of **NS2** scenario with inference extended to four times the training interval. As depicted in Table 4, we can find that the unregularized physics-informed training fails to deliver accurate prediction, underscoring the indispensability of all regularization methods for achieving optimal performance. Removing the alignment regularization $R_A$ leads to a significant drop in performance on long-range prediction, demonstrating its crucial contribution to the model's temporal extrapolation ability. Notably, without smoothing regularization $R_S$, the model struggles to converge to an acceptable level of performance, highlighting the necessity of learning simple latent dynamics for stable and effective training.

Table 4: Ablations on regularization methods. We report $L_2$ relative error (%) within i-th time interval $[(i-1)\Delta T, i\Delta T)$, where $i \in \{1, 2, 3, 4\}$ and $\Delta T = T_{tr}$. The default setting marked in gray.

| $R_A$ | $R_S$ | 1ST$\Delta T$ | 2ND$\Delta T$ | 3RD$\Delta T$ | 4TH$\Delta T$ |
|---|---|---|---|---|---|
| ✓ | ✓ | **4.59** | **10.02** | **14.90** | **19.57** |
| ✗ | ✓ | 4.76 | 11.00 | 17.02 | 24.53 |
| ✗ | ✗ | 17.90 | 33.83 | 43.98 | 51.87 |

### 4.3 DOWNSTREAM TASKS

Having established PIDO's robustness to unseen initial conditions and PDE coefficients, we now explore its representation transferability to downstream tasks. Ideally, transferable representations

Table 5: Downstream tasks on NS equations. For long-term integration, we report the accumulated $L_2$ relative error (%) in $[0, i\Delta T)$, where $i \in \{1, 4, 7, 10\}$ and $\Delta T = T_{tr}$. For inverse problem, we report the $L_2$ relative error (%) of the predicted PDE coefficients under different snapshots N. We compare different training settings of PIDO, including training from scratch (FS), finetuning the dynamics model of pretrained PIDO (FT-DYN) and finetuning its all components (FT-ALL).

| | (a) Long-term integration | | | | | | (b) Inverse problem | | | |
|---|---|---|---|---|---|---|---|---|---|---|
| | PI-DEEPONET | | PIDO | | | | PINN | PIDO | | |
| | FS | FT | FS | FT-DYN | FT-ALL | | FS | FS | FT-DYN | FT-ALL |
| $1\Delta T$ | 8.85 | 5.95 | 1.01 | 1.52 | **0.53** | N=10 | 2.69 | 1.34 | 0.17 | **0.07** |
| $4\Delta T$ | 57.85 | 20.31 | 13.50 | 5.15 | **2.38** | N=5 | 3.38 | 2.22 | 0.31 | **0.21** |
| $7\Delta T$ | 64.14 | 26.11 | 29.11 | 7.71 | **3.99** | N=3 | 8.99 | 9.54 | 2.44 | **1.80** |
| $10\Delta T$ | 68.45 | 24.68 | 36.53 | 8.33 | **5.75** | N=2 | 15.47 | 14.39 | 3.63 | **2.92** |

should simplify learning for subsequent problems. Here, we investigate how a PIDO pre-trained on the **NS2** scenario performs on downstream tasks like long-term integration and inverse problems.

**Long-term Integration.** Training a physics-informed neural PDE solver for long temporal horizons presents a significant challenge. We adopt the training setting outlined in Wang & Perdikaris (2023), which reformulates the long-term integration problem as a series of initial value problems solved within a shorter horizon ($T_{tr}$). Data snapshots are uniformly sampled across the entire test horizon ($T_{ts}$) to serve as training initial conditions. The model, trained on the shorter interval $T_{tr}$, can iteratively predict future states by using previous prediction at the end of $T_{tr}$ as new initial condition, extending its effective horizon without direct long-term training. While this approach can be sensitive to the number of training initial conditions, PIDO's pre-training knowledge mitigate this limitation.

We consider a challenging setting where the test horizon $T_{ts}$ is ten times longer than the training horizon $T_{tr}$. We set $T_{tr}$ to 5s. The data is generated with a Reynolds number $\alpha = 950$, which is unseen by PIDO during the pre-training stage. Ten data snapshots are uniformly sampled from the test horizon as training initial conditions. We compare PIDO against PI-DeepONet. When inference, PIDO can directly predict the entire horizon, while PI-DeepONet relies on the iterative scheme. As shown in Table 5a, both PI-DeepONet and PIDO struggle to achieve satisfactory long-term performances when trained from scratch. In contrast, the pre-trained PIDO exhibits significant improvement (77% error reduction) through fine-tuning the dynamics model. This showcases its effectiveness in enhancing long-term predictions with insufficient data. Furthermore, fine-tuning all components of the pre-trained PIDO can lead to further performance gains.

**Inverse Problem.** PINNs have demonstrated efficacy in solving inverse problems (Raissi et al., 2019; 2020), aiming to recover the PDE coefficients $\boldsymbol{\alpha}$ from a limited set of observations (Pakravan et al., 2021; Zhao et al., 2022; Nair et al., 2023). We then showcase the effectiveness of PIDO's pretrained knowledge in this context by comparing it against PINN. We follow Raissi et al. (2019) to treat coefficients $\boldsymbol{\alpha}$ as learnable parameters and optimize them with a neural network to simultaneously fit the observed data and satisfy the PDE constraints. We focus on a case with Reynolds number $\alpha = 950$ and a training horizon of $T_{tr} = 10s$. Training data consists of N solution snapshots uniformly sampled across the entire horizon. For each snapshot, 5% of spatial locations are randomly selected as observed data points. Table 5b shows that PIDO consistently outperforms its from-scratch counterpart when only finetuning the dynamics model. Notably, even with only two snapshots, the pretrained PIDO achieves accurate predictions. This remarkable performance highlights the capability of PIDO's transferable representations in alleviating the data scarcity burden in inverse problems.

## 5 CONCLUSION

In this paper, we propose PIDO, a novel physics-informed neural PDE solver demonstrating exceptional generalization across diverse PDE configurations. PIDO effectively leverages the shared structure of dynamical systems by projecting solutions into a latent space and learning their dynamics conditioned on PDE coefficients. To tackle the challenges of physics-informed dynamics modeling, we adopt an innovative perspective by diagnosing and mitigating them in latent space, resulting in a significant improvement in the model's temporal extrapolation and training stability. Extensive experiments on 1D and 2D benchmarks demonstrate PIDO's generalization ability to initial conditions, PDE coefficients and training time horizons, along with transferability to downstream tasks.

## REPRODUCIBILITY STATEMENT

In Section 4.1, we detail the dataset generation process used for our benchmarks. The experimental configurations are described in Appendix C. Pseudo-code outlining the training and testing procedures can be found in Algorithm 1 and Algorithm 2, respectively. Additionally, our implementation code is provided in the supplementary materials.

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

# A ADDITIONAL RESULTS

## A.1 FULL RESULTS ON 1D AND 2D BENCHMARKS

We present performance of all methods on the training and test sets in Table 6.

Table 6: Full results on 1D and 2D benchmarks. We report the $L_2$ relative error (%) over the training horizon (IN-T) and the subsequent duration (OUT-T). The best results are **bold-faced**.

| | CE1 | | CE2 | | CE3 | | NS1 | | NS2 | |
|---|---|---|---|---|---|---|---|---|---|---|
| MODEL | IN-T | OUT-T | IN-T | OUT-T | IN-T | OUT-T | IN-T | OUT-T | IN-T | OUT-T |
| *Training set* | | | | | | | | | | |
| PI-DEEPONET | 4.08 | 8.26 | 16.87 | 36.63 | 7.20 | 15.54 | 20.41 | 34.98 | 25.43 | 45.26 |
| PINODE | 9.41 | 22.14 | 9.66 | 26.10 | 17.47 | 37.37 | 14.50 | 49.73 | 15.10 | 45.83 |
| MAD | 2.19 | 4.87 | 7.77 | 21.66 | 3.94 | 9.43 | 12.93 | 28.67 | 14.92 | 30.83 |
| PIDO | **1.38** | **1.99** | **2.33** | **5.08** | **3.12** | **5.40** | **1.81** | **4.10** | **3.20** | **7.04** |
| *Test set* | | | | | | | | | | |
| PI-DEEPONET | 4.18 | 8.61 | 17.17 | 36.16 | 7.57 | 15.74 | 23.57 | 36.10 | 29.86 | 47.10 |
| PINODE | 10.44 | 24.75 | 11.03 | 28.69 | 18.21 | 39.41 | 16.44 | 53.52 | 17.56 | 46.67 |
| MAD | 3.98 | 9.32 | 12.00 | 27.97 | 6.78 | 17.10 | 14.85 | 30.50 | 16.95 | 33.49 |
| PIDO | **1.48** | **2.24** | **3.02** | **7.15** | **3.19** | **8.08** | **2.35** | **5.43** | **4.59** | **10.02** |

## A.2 EXTRAPOLATION OUTSIDE THE TRAINING DISTRIBUTION OF PDE COEFFICIENTS.

We examine the extrapolation capability of PIDO beyond the training distribution of Reynolds numbers in the **NS2** setting. We focused on higher Reynolds numbers as they represent more complex fluid dynamics. Our comparisons with MAD, the best-performing baseline, are detailed in Table 7.

Table 7: Extrapolation outside the training distribution of Reynolds number in **NS2** setting. We report the $L_2$ relative error (%) over the training horizon (IN-T) and the subsequent duration (OUT-T).

| MODEL | $\alpha = 550$ | $\alpha = 650$ | $\alpha = 1450$ | $\alpha = 1550$ | $\alpha = 1650$ | $\alpha = 1750$ | $\alpha = 1850$ |
|---|---|---|---|---|---|---|---|
| *In-t* | | | | | | | |
| MAD | 1.83 | 4.54 | 24.98 | 25.81 | 27.57 | 28.90 | 29.91 |
| PIDO | 1.08 | 0.68 | 10.77 | 12.17 | 13.89 | 15.04 | 16.51 |
| *Out-t* | | | | | | | |
| MAD | 3.71 | 10.59 | 46.63 | 49.50 | 53.37 | 55.56 | 56.72 |
| PIDO | 3.38 | 2.11 | 22.32 | 25.90 | 30.29 | 32.90 | 36.60 |

## A.3 COMPARISONS WITH DATA-DRIVEN BASELINES

In addition to DINo, we compare our method with other data-driven approaches, including FNO (Li et al., 2020b) and DeepONet (Lu et al., 2021a), in Table 8. Our results show that PIDO consistently outperforms both FNO and DeepONet in the *In-t* and *Out-t* settings.

We also compare our method with PINO (Li et al., 2021), which approximates the PDE-based loss using the finite difference method, making it sensitive to the time step size. To ensure training stability, we adopt a time step size of 0.2 seconds for PINO, which is five times smaller than that used for PIDO. Our results indicate that PIDo demonstrates competitive performance with PINO in the *In-t* prediction but outperforms it in the *Out-t* scenario.

## A.4 ABLATION ON INR ARCHITECTURES

We investigate the impact of different choices for the INR architectures within PI-DeepONet, MAD, and our proposed method in Table 9. The results demonstrate that employing FourierNets consistently

Table 8: Comparison with data-driven methods. $L_2$ relative error in **NS1** is reported (%).

| | | TRAIN | | TEST | |
| MODEL | DATASET | IN-T | OUT-T | IN-T | OUT-T |
|---|---|---|---|---|---|
| PINO | - | 3.88 | 10.11 | 3.89 | 10.16 |
| PIDO | - | 1.81 | 4.10 | **2.35** | **5.43** |
| DEEPONET | 100% | 7.15 | 12.23 | 10.28 | 13.55 |
| FNO | 100% | 2.83 | 8.74 | 2.86 | 8.83 |
| DINO | 100% | **1.42** | **3.32** | 4.26 | 5.73 |

improves the performance of all three methods on both *In-t* and *Out-t* metrics compared to using MLPs with tanh or sin activation functions. Furthermore, our method achieves superior performance over both PI-DeepONet and MAD in all evaluated INR settings.

Table 9: Ablation on INR architectures. We report the $L_2$ relative error (%) over the training horizon (IN-T) and the subsequent duration (OUT-T) in CE1 scenarios. The best results are **bold-faced**.

| INR | PI-DEEPONET | | MAD | | PIDO | |
|---|---|---|---|---|---|---|
| | IN-T | OUT-T | IN-T | OUT-T | IN-T | OUT-T |
| MLP (*tanh*) | 15.25 | 27.72 | 4.24 | 17.06 | 2.66 | 6.40 |
| MLP (*sin*) | 17.75 | 26.67 | 4.14 | 15.98 | 2.33 | 5.59 |
| FOURIERNET | **4.18** | **8.61** | **3.98** | **9.32** | **1.48** | **2.24** |

## A.5 SAMPLE EFFICIENCY

We conduct a comparative analysis of PIDO against baseline models across varying numbers of training pairs. Specifically, we focus on the **CE3** scenario and sample subsets of training pairs with different ratios, denoted as $s \in \{12.5\%, 25\%, 50\%, 100\%\}$, where $s = 100\%$ corresponds to the complete training set. We report results in the test set *In-t* in Figure 6. Our observations indicate that PIDO consistently achieves optimal performance across all sample ratios and exhibits reduced sensitivity to the reduction of training pairs in comparison to PI-DeepONet. Notably, PIDO, even when utilizing only 12.5% of training pairs, achieves comparable performance with the other two baselines employing 100% of the training pairs.

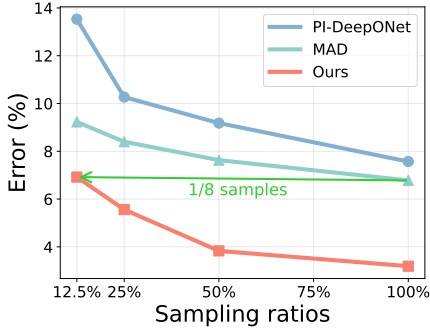

Figure 6: **CE3** test *In-t* performance vs. numbers of training pairs.

## A.6 1D COMBINED EQUATIONS

We provide in Figure 7 visualizations of PIDO in **CE3** test set.

## A.7 2D NAVIER-STOKES EQUATIONS

We provide in Figure 8 visualizations of PIDO in **NS2** test set.

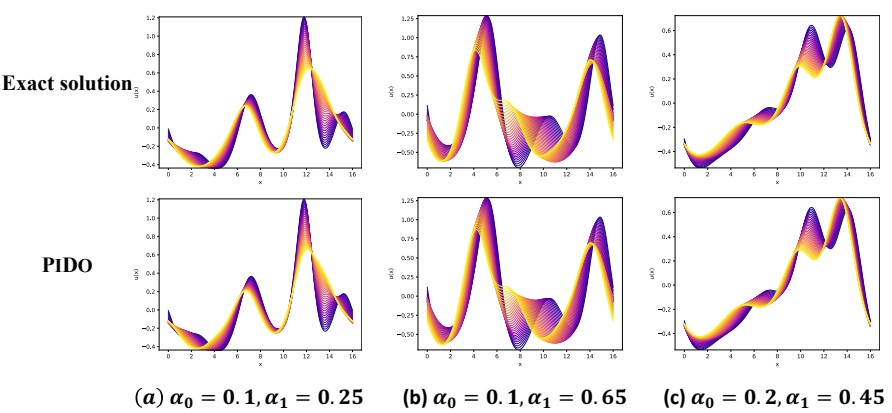

(a) $\alpha_0 = 0.1, \alpha_1 = 0.25$     (b) $\alpha_0 = 0.1, \alpha_1 = 0.65$     (c) $\alpha_0 = 0.2, \alpha_1 = 0.45$

Figure 7: Prediction of PIDO on 1D combined equations with different $\alpha$.

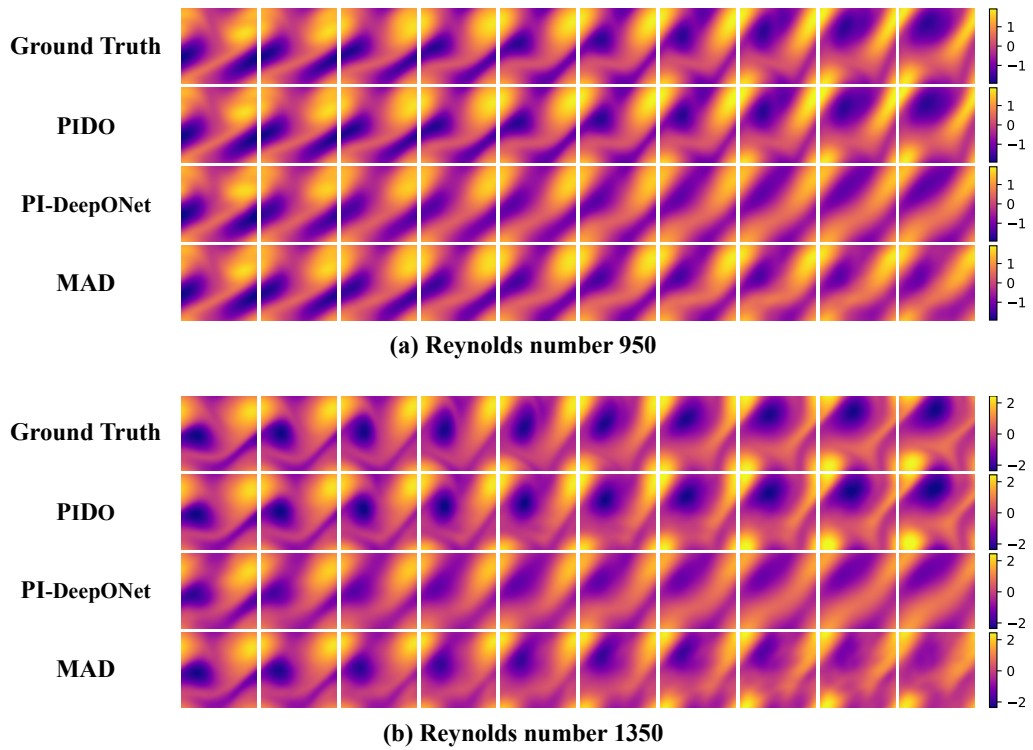

Figure 8: Prediction of PIDO on 2D NS equations with different Reynolds number. The last 5 frames are beyond the training horizon.

## A.8 LONG-TERM INTEGRATION

We provide in Figure 9 visualizations of PIDO (FS) and PIDO (FT) in the long-term integration task.

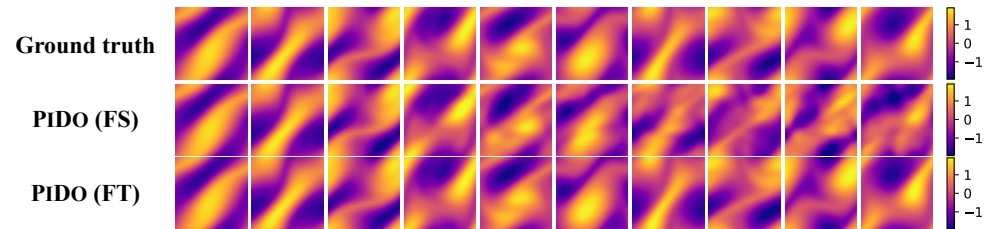

Figure 9: Long-term integration starting from t=0s to t=45s at a step of 5s.

## B  AUTO-DECODER VERSUS AUTO-ENCODER FOR PHYSICS-INFORMED EDM

Since physics-informed training strategies for auto-encoders differ significantly from those used for auto-decoders (our method), this section discusses these key differences and details the specific training settings employed in auto-encoder methods.

Previous work PINODE (Sholokhov et al., 2023) utilizes an auto-encoder framework for physics-informed training. Unlike our auto-decoder, which takes spatial coordinates $x$ and embeddings $c$ as input data and output predictions of states $u$, PINODE's encoder $\mathcal{E}$ operates in the opposite direction. It takes $u$ as input and output $c = \mathcal{E}(u)$. This allows PINODE to calculate the derivative of $c$ w.r.t $u$ using Auto-Differentiation (AD). Consequently, PINODE can derive the temporal derivative of its embedding $c$, assuming the input data $u$ follows the PDE $\frac{\partial u}{\partial t} = \mathcal{L}(u)$:

$$\frac{\partial c}{\partial t} = \frac{\partial c}{\partial u} \cdot \frac{\partial u}{\partial t} = \frac{\partial c}{\partial u} \cdot (-\mathcal{L}(u)). \tag{17}$$

This derivative then serves as labels to train the dynamics model $\mathcal{F}$ by the following loss:

$$l_{\text{PDE}}(\theta_{\mathcal{E}}, \theta_{\mathcal{F}}) = \left\| \frac{\partial c}{\partial t} - \mathcal{F}(c) \right\|_2^2 = \left\| \frac{\partial c}{\partial u} \cdot (-\mathcal{L}(u)) - \mathcal{F}(c) \right\|_2^2, \tag{18}$$

where $\theta_{\mathcal{E}}$ and $\theta_{\mathcal{F}}$ denote parameters of $\mathcal{E}$ and $\mathcal{F}$, respectively.

However, this approach has three limitations. First, PINODE's encoder and decoder adheres to a fixed grid for both input data and output predictions, limiting its flexibility. Second, PINODE relies on an analytical representation of the input data $u$ to compute its spatial derivatives involved in $\mathcal{L}(u)$ in Equation (18). This is achieved by pre-defining an analytical distribution from which the input data is sampled. Finally, PINODE assumes the input data distribution accurately reflects the true PDE solutions. However, finding such a representative distribution in real-world scenarios can be challenging.

Deviations from this assumption can significantly degrade PINODE's performance. Our experiments (Table 10) demonstrate this sensitivity. In these experiments, we sample PINODE's input data from exact PDE solutions (ideal scenario) or the distribution of initial conditions (the only data we can access in the data-constrained scenario). The results show that PINODE's time extrapolation performance suffers significantly when the input data deviates from the true distribution of PDE solutions. Additionally, increasing the number of initial conditions offered little improvement, further highlighting PINODE's dependence on a suitable input data distribution. Given the data-constrained setting of our experiments in Section 4, we employ the distribution of initial conditions as input data for PINODE to ensure a fair comparison with other methods.

In contrast, PIDO adopts the auto-decoder framework, which is grid-independent and enables the calculation of spatial derivatives of predicted solutions through AD. Moreover, the training of PIDO does not require any prior knowledge about data distribution, making it more robust for real-world scenarios.

## C  EXPERIMENTS SETUPS

### C.1  TRAINING DETAILS

**1D combined equations.**  For this problem, we consider the periodic boundary condition and construct training and test sets with initial conditions sampling from the super-position of sinusoidal

Table 10: The performance of PINODE with different input data. $N_{\text{IC}}$ denotes the number of initial conditions. We report the $L_2$ relative error (%) on the CE3 scenario.

| METHOD | INPUT DATA | IN-T | OUT-T |
|--------|-----------|------|-------|
| PINODE | EXACT SOLUTIONS | 5.77 | 11.02 |
| PINODE | INITIAL CONDITIONS ($N_{\text{IC}}$=3584) | 18.21 | 39.41 |
| PINODE | INITIAL CONDITIONS ($N_{\text{IC}}$=7168) | 18.25 | 41.26 |
| PINODE | INITIAL CONDITIONS ($N_{\text{IC}}$=14336) | 17.74 | 39.19 |
| PIDO | INITIAL CONDITIONS ($N_{\text{IC}}$=3584) | 3.19 | 8.08 |

waves given by $\sum_{i=1}^{N} A_i \sin(\frac{2\pi k_i}{L_x}x + b_i)$, where $\{A_i\}$, $\{b_i\}$ and $\{k_i\}$ denote random amplitudes, phases and integer wave numbers. We set $L_x = 16$, $T_{tr} = 0.96s$ and $T = 1.92s$. We employ a uniform spatial discretization of 400 cells encompassing the interval [0, 16). The temporal domain is discretized into 60 time steps using a uniform spacing over the interval [0, 1.92]. During training, we sample collocation points from this grid for physics-informed training. Each model is trained for 3000 epochs with a batch size of 128 and a learning rate of 1e-3. For PIDO, the weights of alignment and smoothing regularization are set to 1 and 0.01, respectively. We consider three scenarios for this equation:

- **CE1** for Burgers' equation with $\boldsymbol{\alpha} = (0.1, 0)$, generating initial conditions with $A_i \in [-0.5, 0.5]$, $k_i \in \{1, 2\}$, $b_i \in [0, 2\pi]$ and N=2; We generate 3584 trajectories for training and 512 trajectories for testing.

- **CE2** for KdV equation with $\boldsymbol{\alpha} = (0, 0.05)$, generating initial conditions with $A_i \in [-0.5, 0.5]$, $k_i \in \{1, 2\}$, $b_i \in [0, 2\pi]$ and N=2; We generate 3584 trajectories for training and 512 trajectories for testing.

- **CE3** for combined equation with $\boldsymbol{\alpha} \in \{(\alpha_0, \alpha_1) | 0 < \alpha_0 \leq 0.4, 0 < \alpha_1 \leq 0.65\}$. Specifically, we use the training set $\boldsymbol{\alpha}_{tr} \in \{0.1, 0.2, 0.3, 0.4\} \times \{0.05, 0.25, 0.45, 0.65\}$ and the test set $\boldsymbol{\alpha}_{ts} \in \{0.15, 0.25, 0.35\} \times \{0.15, 0.35, 0.55\}$. We generate initial conditions with $A_i \in [0, 1]$, $k_i \in \{1, 2\}$, $b_i \in [0, 2\pi]$ and N=2; We generate 224 trajectories for each configuration of training coefficients (3584 in total) and 32 trajectories for each configuration of testing coefficients (288 in total).

**2D Navier-Stokes equation.** For this problem, trajectories are simulated under periodic boundary conditions, employing initial conditions described in Li et al. (2020b). We set $T_{tr} = 5s$ and $T = 10s$. We employ a uniform spatial discretization of 64*64 cells encompassing the interval $[0, 1)^2$. We consider a temporal domain of [0, 10] and use a time step size of 0.5 seconds for PI-DeepONet and MAD to ensure training stability. For PIDO, we use a time step size of 1 second, benefiting from the latent dynamics smoothing. We sample collocation points from this grid for physics-informed training. For PIDO, the weights of alignment and smoothing regularization are set to 1 and 0.01, respectively. We consider two scenarios:

- **NS1** for fixed Reynolds number $\alpha = 1000$. We generate 1024 trajectories for training and 128 trajectories for testing. Each model is trained for 3000 epochs with a batch size of 16 and a learning rate of 2e-3. For the dynamics model of PIDO, the learning rate is set to 2e-4.

- **NS2** for diverse Reynolds numbers. We utilizes a training set encompassing $\alpha$ values sampled from $\{700, 800, 900, 1000, 1100, 1200, 1300, 1400\}$ and a testing set incorporating Reynolds numbers from $\{750, 850, 950, 1050, 1150, 1250, 1350\}$. We generate 256 trajectories for each configuration of training coefficients and 32 trajectories for each configuration of testing coefficients. Each model is trained for 6000 epochs with a batch size of 64 and a learning rate of 1e-3. For the dynamics model of PIDO, the learning rate is set to 1e-4.

## C.2 IMPLEMENTATIONS

**PIDO.** The decoder is a FourierNet (Fathony et al., 2020) with 3 hidden layers and a width of 64. We opted to employ FourierNet due to its demonstrated superior performance in tasks similar to ours. To maintain a fair and controlled comparison with baseline methods, we utilize the same network

Table 11: The computational time and memory usage of each method.

| METHOD | TIME PER EPOCH (SECOND) | MEMORY PER SAMPLE (MB) |
|---|---|---|
| PI-DEEPONET | 27 | 533 |
| MAD | 29 | 569 |
| PIDO | 35 | 582 |

architecture across all approaches in this work. An FourierNet with k hidden layers is defined via the following recursion

$$\boldsymbol{z}^{(1)} = \sin(\omega^{(1)}\boldsymbol{x} + \phi^{(1)}), \boldsymbol{z}^{(i+1)} = (W^{(i)}\boldsymbol{z}^{(i)} + b^{(i)}) \circ \sin(\omega^{(i+1)}\boldsymbol{x} + \phi^{(i+1)}), \ i = 1, 2, ..., k,$$

$$\boldsymbol{z}_{\text{out}} = W^{(k+1)}\boldsymbol{z}^{(k+1)} + b^{(k+1)},$$

where $\boldsymbol{x}$ is the input coordinates, $\circ$ is the elemental multiplication and $\{W^{(i)}, b^{(i)}, \omega^{(i)}, \phi^{(i)}\}$ denote the trainable parameters. To incorporate embedding $\boldsymbol{c}$ into a FourierNet, we modulate both the amplitudes and phases of the sinusoidal waves generated by the hidden layers:

$$\boldsymbol{z}^{(i+1)} = (W^{(i)}\boldsymbol{z}^{(i)} + b^{(i)} + W_A^{(i)}\boldsymbol{c}) \circ \sin(\omega^{(i+1)}\boldsymbol{x} + \phi^{(i+1)} + W_P^{(i)}\boldsymbol{c})), \ i = 1, 2, ..., k.$$

The dynamics model is a 4-layer MLP with a width of 512. The activation function of dynamics model is Swish. We use the RK4 integrator via TorchDiffEq (Chen, 2018) for the training of dynamics model. We set the code size to 64 for 1D combined equations and to 128 for 2D NS equations.

**PI-DeepONet.** The trunk net is a FourierNet with 3 hidden layers and a width of 64. The branch net is a 4-layer SIREN (Sitzmann et al., 2020) with a width of 512.

**PINODE.** Both the encoder and decoder are SIRENs with 3 hidden layers and a width of 64. The encoder takes the discrete initial conditions as input, which are vectors of 400 elements for 1D problems and 4096 elements (64*64) for 2D problems, and generates the latent embedding. The decoder operates in the opposite direction. The dynamics model and latent code configurations are identical to those employed in our PIDO method.

**MAD.** The decoder is a FourierNet with 3 hidden layers and a width of 64. We set the embedding size to 64 for 1D combined equations and to 128 for 2D NS equations. When inference on new initial conditions or PDE coefficients, we only finetune the learnable embeddings while freeze the parameters of decoder.

### C.3 COMPUTATIONAL EFFICIENCY

We used 4 NVIDIA RTX3090 GPUs for all experiments. We compare the computational time and memory usage of PIDO and the baselines in the CE1 setting in Table 11. While PIDO incurs slightly higher memory and computation demands due to its autoregressive Neural ODE architecture, this trade-off is demonstrably worthwhile. The resulting performance boost is significant, and overall resource requirements remain relatively low.

## D  LIMITATIONS AND FUTURE WORK

In this work, we mainly focus on the coefficient-aware dynamics modeling, concerning the generalization w.r.t initial conditions, PDE coefficients and time horizons. However, we acknowledge two key limitations that motivate future research directions: First, while we employ periodic boundary conditions in all our experiments, a crucial future direction is to extend the generalization study to handle diverse boundary conditions and geometries. A promising approach might involve utilizing multiple decoders, each tailored to specific boundary conditions. Second, the smoothing regularization used for stability training can potentially penalize high-frequency physics information, which is crucial for accurate modeling. Currently, we achieve a trade-off between stability and accuracy with a suitable regularization weight. It is a promising avenue to investigate alternative regularization techniques that improve stability without compromising the capture of high-frequency details.

---

**Algorithm 1** Training Procedure

---

**Input:** Decoder parameters $\boldsymbol{\theta}_D$, NODE parameters $\boldsymbol{\theta}_{\mathcal{F}}$, initial conditions and PDE coefficients $(\boldsymbol{\phi}^i, \boldsymbol{\alpha}^i)$, initial embeddings $(\boldsymbol{c}_0^i)$, latent embeddings for consistency regularization $(\{\bar{\boldsymbol{c}}_t^i\}_{t=1}^N)$

**Initialize:** set embeddings to zero $\boldsymbol{c}_0^i \leftarrow 0, \forall i$ and $\bar{\boldsymbol{c}}_t^i \leftarrow 0, \forall (i, t)$

**Hyper-parameters:** learning rates for latent embeddings $\lambda_c$, Decoder $\lambda_D$ and NODE $\lambda_{\mathcal{F}}$;

**repeat**

    Sample one pair of $(\boldsymbol{\phi}^i, \boldsymbol{\alpha}^i, \boldsymbol{c}_0^i, \{\bar{\boldsymbol{c}}_t^i\}_{t=1}^N)$;

    $\{\boldsymbol{c}_t^i\}_{t=1}^N \leftarrow \mathcal{F}(\boldsymbol{c}_0^i, \boldsymbol{\alpha}^i | \boldsymbol{\theta}_{\mathcal{F}})$;       // unroll the trajectory

    $\{\boldsymbol{u}_t^i\}_{t=0}^N \leftarrow \mathcal{D}(\{\boldsymbol{c}_t^i\}_{t=0}^N | \boldsymbol{\theta}_D)$;     // obtain predicted solutions

    $\{\bar{\boldsymbol{u}}_t^i\}_{t=1}^N \leftarrow \mathcal{D}(\{\bar{\boldsymbol{c}}_t^i\}_{t=1}^N | \boldsymbol{\theta}_D)$;     // prepare for the auto-decoding of predicted solutions

    $/*$ update initial embeddings $*/$

    $\boldsymbol{c}_0^i \leftarrow \boldsymbol{c}_0^i - \lambda_c \nabla_{\boldsymbol{c}_0^i} l_{\text{IC}}(\boldsymbol{\phi}_i, \boldsymbol{u}_0^i)$;

    $/*$ update network parameters with physics-informed loss and regularization $*/$

    $\boldsymbol{\theta}_D \leftarrow \boldsymbol{\theta}_D - \lambda_D \nabla_{\boldsymbol{\theta}_D} (l_{\text{IC}}(\boldsymbol{\phi}_i, \boldsymbol{u}_0^i) + l_{\text{BC}}(\{\boldsymbol{u}_t^i\}_{t=0}^N) + l_{\text{PDE}}(\{\boldsymbol{u}_t^i\}_{t=0}^N))$;

    $\boldsymbol{\theta}_{\mathcal{F}} \leftarrow \boldsymbol{\theta}_{\mathcal{F}} - \lambda_{\mathcal{F}} \nabla_{\boldsymbol{\theta}_{\mathcal{F}}} (l_{\text{IC}}(\boldsymbol{\phi}_i, \boldsymbol{u}_0^i) + l_{\text{BC}}(\{\boldsymbol{u}_t^i\}_{t=0}^N) + l_{\text{PDE}}(\{\boldsymbol{u}_t^i\}_{t=0}^N) + R_C(\{\boldsymbol{c}_t^i\}_{t=1}^N, \{\bar{\boldsymbol{c}}_t^i\}_{t=1}^N) + R_S(\{\boldsymbol{c}_t^i\}_{t=1}^N))$;

    $/*$ update latent embeddings for consistency regularization $*/$

    $l_{\bar{\boldsymbol{c}}_t} \leftarrow \mathbb{E}_{\boldsymbol{x} \in \Omega} \|\bar{\boldsymbol{u}}_t^i(\boldsymbol{x}) - \boldsymbol{u}_t^i(\boldsymbol{x})\|_2^2, \forall t$;

    $\bar{\boldsymbol{c}}_t^i \leftarrow \bar{\boldsymbol{c}}_t^i - \lambda_c \nabla_{\bar{\boldsymbol{c}}_t^i} l_{\bar{\boldsymbol{c}}_t}, \forall t$ ;     // auto-decode the predicted solutions

**until** convergence

---

**Algorithm 2** Testing Procedure

---

**Input:** Decoder $\boldsymbol{\theta}_D$, NODE $\boldsymbol{\theta}_{\mathcal{F}}$, initial conditions and PDE coefficients $(\boldsymbol{\phi}, \boldsymbol{\alpha})$, initial embeddings $(\boldsymbol{c}_0)$

**Initialize:** set embeddings to zero $\boldsymbol{c}_0 \leftarrow 0$

**Hyper-parameters:** learning rates for latent embeddings $\lambda_c$, optimization step for auto-decoding $S$;

**for** $s = 1$ **to** $S$ **do**

    $\boldsymbol{u}_0 \leftarrow \mathcal{D}(\boldsymbol{c}_0 | \boldsymbol{\theta}_D)$;

    $l_{\boldsymbol{c}_0} \leftarrow \mathbb{E}_{\boldsymbol{x} \in \Omega} \|\boldsymbol{\phi}(\boldsymbol{x}) - \boldsymbol{u}_0(\boldsymbol{x})\|_2^2$;

    $\boldsymbol{c}_0 \leftarrow \boldsymbol{c}_0 - \lambda_c \nabla_{\boldsymbol{c}_0} l_{\boldsymbol{c}_0}$;     // auto-decode

**end for**

$\{\boldsymbol{c}_t\}_{t=1}^N \leftarrow \mathcal{F}(\boldsymbol{c}_0, \boldsymbol{\alpha} | \boldsymbol{\theta}_{\mathcal{F}})$;     // unroll the trajectory

$\{\boldsymbol{u}_t\}_{t=0}^N \leftarrow \mathcal{D}(\{\boldsymbol{c}_t\}_{t=0}^N | \boldsymbol{\theta}_D)$;     // obtain predicted solutions

---

# E   BROADER IMPACTS

This work may inherit both the positive and negative impacts associated with deep learning-based PDE solvers. On the positive side, our work has the potential to significantly reduce the time and resources required for simulations and modeling in fields such as aerodynamics, material science, and fluid dynamics, thereby accelerating innovation cycles. Conversely, on the negative side, our work may inherit biases from the training configuration, which could lead to difficulties in applying the model to real-world problems with different underlying conditions.

# F   LICENSES OF ASSETS

DeepXDE library (Lu et al., 2021c) is under the LGPL-2.1 License (Lu et al., 2021b). FourierNet (Fathony et al., 2020) is open-sourced under the AGPL-3.0 license (Fathony et al., 2021). Torchdiffeq is under the MIT license (Chen, 2018).

