# OpenReview forum: "Physics-informed Dynamics Representation Learning for Parametric PDEs"
_ICLR.cc/2025/Conference — ICLR 2025 Conference Withdrawn Submission_

### Official Review · Reviewer_WG6d · 2024-10-20

**Soundness:** 3
**Presentation:** 3
**Contribution:** 3
**Rating:** 6
**Confidence:** 3

**Summary:**

This paper introduces a novel physics-informed neural PDE solver, PIDO, which projects the PDE solutions into a latent space via auto-decoding and subsequently learns the dynamics of these latent embeddings conditioned on the PDE coefficients. Optimization challenges associated with physics-informed losses are diagnosed in the latent space, and latent dynamics smoothing and alignment are introduced to mitigate them. The effectiveness of the proposed approach is validated on diverse benchmarks, especially for extrapolation and generalization beyond the time horizon experienced during training

**Strengths:**

- Well-written paper overall
- Extensive numerical experiments and ablation studies, with several baselines for comparisons
- Strong results
- Refinements and improvements on the original idea with theoretical justifications

**Weaknesses:**

**Main concern**
- Table 2 and Figure 5: I am a little bit surprised about how bad the L2 relative errors are for the baselines on certain benchmarks, especially on the training horizon. It makes it hard to believe that these are the state-of-the-art existing approaches. Could you specify the hyperparameter tuning strategies used for the different approaches? In particular, could you confirm that you have spent as much effort tuning the hyperparameters of the baselines and of your approach? There might also be other existing better-performing baselines to compare to in addition to the ones presented in the paper. Any additional information that could help understand why it is not surprising that the baselines are performing so badly would be welcome

**Minor weaknesses or suggestions/questions**:
- Figure 1 is not very clear. I think it needs to be rethought to propose a clearer description of the proposed approach
- Figure 2: I am not sure I understand exactly the green and red markings. Is it supposed to show that training is done at the integer times {0,1,2,3,...} but that you also predicted at the in-between value 0.5? If that's the case, I think this could be presented differently in a cleaner way
- Section 3.4.1, 2nd paragraph: I think this needs to be investigated a little bit more. I don't think the results in Figure 2 are sufficient to make these strong conclusions. That is only for one example of dynamical system where the evolution is very slow over the prediction time interval. I am not convinced this applies more systematically, that the model learns excessively fluctuating latent dynamics, and would be curious to see the same experiments carried for faster-varying dynamics. I think this connects as well to the comment made in Appendix D that the smoothing regularization used for stability training can potentially penalize high-frequency physics information. While the latent dynamics smoothing helps for the slower-varying systems, it would be interesting to discuss it in the context of faster-varying systems.
- Section 3.4.2, 1st paragraph: "by analyzing the evolution of latent embeddings over extended time horizons, we trace this issue to
the phenomenon of latent embedding drift". Can you provide more details here about the latent embedding drift, and in particular about the analysis you carried that allowed you to trace the issue back to that phenomenon?
- Figure 4 is not clear. I think the paper needs a better figure to explain the latent space smoothing and the latent space alignment.
- Section 4.2, Generalizing across initial conditions: where are these percentages coming from? I don't see any such results in the tables and figures. Also, it should be made clearer what the percentages mean: what does 63% means, does it mean a l2 relative error going down from 70% to 7% for instance?
- Appendix C.3: The short comment about the computational efficiency should be made in the main text (while table 11 can be kept in the appendix). The results are only provided for the CE1 setting. Could you confirm that similar discrepancies in computational times were observed with the other problems and strategies tested? How does it scale up compared to the other strategies?
- Appendix D about limitations and future work should be in the main conclusion, not in the appendix.

**Questions:**

Suggestions and questions were made in the weaknesses section.

Overall, I think this is a good paper and I would recommend to accept. If the figures describing the approach can be made clearer and my concern about the baselines alleviated, I would be happy to upgrade my score to an 8.

---

### Official Review · Reviewer_Hqy9 · 2024-10-29

**Soundness:** 2
**Presentation:** 3
**Contribution:** 2
**Rating:** 5
**Confidence:** 2

**Summary:**

The paper introduces physics-informed dynamics representation with two regularization techniques to achieve a generalizable learning model to solve PDEs with various configurations. Several experimental tests are conducted by comparing the method with some baselines.

**Strengths:**

The author creatively combines techniques from EDM, NODE, auto-decoding, and smoothing regularization. The design logic for each block is fluent. Several baselines are tested and compared to support the effectiveness of the model.

**Weaknesses:**

Minor concern:

1.	Some shorthands should be explained before use. E.g., in Fig. 1, you may need to explain DEC, ENC, and DYN.

2.	Some typos exist. For example, in Line 175, “n this section”. Line 259, “This provide us”. Please do a complete proofreading.

3.	Why do you conclude that Fig. 2a is a trivial solution caused by large training steps?

4.	What is the performance after the author applies $R_A$ in Fig. 3?

Major concern:

1.	Some notations in Fig. 1 are not explained in the previous pages. So, it’s hard to understand Fig. 1 when reading the caption and descriptions in the Introduction.

2.	The key employed technologies, e.g., auto-decoding and EDM, are not well motivated and explained. Why should we build a latent space? Why should we use auto-decoding?

3.	It seems that smoothing regularization will hurt the expressivity of the model. I suggest that the author conducts a sensitivity analysis with respect to regularization penalty terms and evaluates the performance.

4.	As the author claims that including physics-informed loss is important with limited numbers, the author should give strong numerical support. For example, in Table 3, PIDO and DINO are compared. However, it’s not clear what’s the performance of PIDO under different percentages of datasets. Although there is a sensitivity analysis in Fig. 6 in the Appendix, I don’t see the result of DINO.

5.	In Section 3.3, the author motivates physics-informed loss by saying that the data-driven method requires prohibitive data, which implies that the author’s design can handle data-insufficient cases. However, in Section 3.4.2, the proposed regularization aligns the latent dynamics with the dynamics obtained by data. If data is scarce, the effectiveness of the regularization is doubtful. The author should give some analysis or numerical support.

6.     There is no discussion about the weakness of the model and future work.

**Questions:**

Q1. Please refine the figure and the writing.

Q2. The motivation for designing the latent space and the auto-decoding/EDM should be well-explained.

Q3. Several vague descriptions need to be clarified in my "Minor concerns 3-4".

Q4. The author needs more explanations about the effectiveness of the regularization techniques with numerical support. See my "Major concerns 3-5".

Q5. The author should discuss the weakness of the model and the future work.

---

### Official Review · Reviewer_wEhq · 2024-10-30

**Soundness:** 3
**Presentation:** 3
**Contribution:** 4
**Rating:** 6
**Confidence:** 4

**Summary:**

PINN-based PDE solvers generally suffer from the generalisation to unseen scenarios during its trining, such as unseen PDE parameters and longer time horizon. The paper proposes to introduce latent representation into PINN-based PDE solvers to overcome those limitations. Key idea behind the proposal is that through latent representation, the model can perform the evolution of dynamics in the low dimensional space, which help the model to generalize to unseen scenarios. The model is evaluated with strong PINN-based and data-driven baseline models in challenging out-of-distribution scenarios and shown to outperforms the baselines in majority of the cases.

**Strengths:**

- The paper is generally easy to follow. The motivation of the work is posited very well.

- Incorporating the latent representation into PINN-based PDE simulator is very effective to overcome the limitation of poor generalisation ability of PINN-based simulators. Discussion on the reason for introducing additional regularisation terms is also very convincing providing pre-experimental results. Especially, the issue on falling down into trivial solutions in latent space is one of the fundamental issues when incorporating latent representation into PDE simulators. The authors find a cleaver way to make use of the equations at line 323 to enable incorporating Jacobin/kinetic regularisation technique in the PINN-based training framework. An ablation study for the loss function is also conducted, which strongly supports the need of regularisation terms to achieve strong generalisation to long time horizon and unseen initial conditions.

- Experiments are exhaustive. The methods are tested on scenarios in which initial conditions, PDE coefficients, or training horizon vary. Comparison with the data-driven approach is also conducted and the proposed method is shown to outperforms a baseline in the long-time horizon prediction task which the existing data-driven approaches are generally very good at solving.

**Weaknesses:**

Some description of the training procedure is unclear or misleading.
- How to train encoder and decoder is not clearly explained. The decoder is trained through the equation (13) and this formulation includes $c_{0}^{i}$ which is obtained through $\mathcal{E}$. But this encoder is obtained through equation (6), which assumes one to have a learned decoder. It would be very helpful to provide a step-by-step description of the training procedure, clarifying how the encoder and decoder are initialized and trained iteratively.
- Arguments of the loss functions such as (9) and (10) include both trainable and fixed parameters, which makes it a bit difficult to understand the overall training procedure. Maybe it would be helpful to provide a clear distinction between trainable and fixed parameters in the loss functions, perhaps using different notation or explicitly stating which parameters are optimized in each step of the training process.

Some statistical parameters are missing from the results. While the metric used for experiments is clearly explained, the numbers reported in tables and figures do not have error bars or standard deviations, which makes the results sound less reliable. I encourage the authors to include error bars or standard deviations for the reported results in tables and figures. This would provide a more complete picture of the method's performance and variability.

**Minor:**
- L.175 ‘n this section'
- Numbers in some tables are not in bold fond while they are the best. For example Table 7 and 8.

**Questions:**

See Weakness section.

---

### Official Review · Reviewer_HeQ9 · 2024-11-02

**Soundness:** 2
**Presentation:** 3
**Contribution:** 2
**Rating:** 3
**Confidence:** 5

**Summary:**

This paper presents PIDO, a physics-informed neural PDE solver that exhibits generalization across diverse PDE configurations by leveraging latent space dynamics. It further enhances temporal extrapolation and training stability through regularization techniques within the latent space, resulting in superior performance compared to some existing methods.

**Strengths:**

+ The paper is well-written and easy to follow.
+ Results show the effectiveness of PIDO for specified datasets.
+ Extensive analyses were performed.

**Weaknesses:**

- The datasets used to demonstrate the performance of the model are too simple. More complex examples, e.g., 3D reaction-diffusion system, 3D NS dataset, should be tested. In addition, the 2D NS example in the paper is rather simple, where the flow patterns are simple since the domain $[0, 1]^2$ with periodic BC is too small. The authors should consider a larger domain, e.g., $ [0, 2\pi]^2$ (please see the JAX-CFD paper).
- The comparison with baseline models is weak, where the considered baselines (e.g., PI-DeepONet) are outdated. There exist many physics-informed neural operators in the literature that need to be compared.
- The paper claims in Table 1 that the proposed model is flexible on a given grid choice. However, the example didn’t show any evidence. Testing the generalization of the model over different mesh grids (including unstructured) should be conducted.
- The model should also be tested to solve PDEs with zero training data, when the complete information of PDE, IC, and BC is given.

**Questions:**

Please see the weaknesses above.

---

### Official Review · Reviewer_e7us · 2024-11-02

**Soundness:** 3
**Presentation:** 3
**Contribution:** 2
**Rating:** 6
**Confidence:** 3

**Summary:**

The paper introduces a new framework for learning dynamical systems governed by PDEs called PiDo. It exploits physic-informed loss training and latent space dynamics modeling via auto-decoding. It leverages strengths from data-free approaches (PINNs) and data-driven approaches (EDMs, NOs) to improve neural solvers ability to generalize to different ICs, PDE coefficients, training horizons, while remaining data-free. They notably introduce two regularization losses for stabilizing the training process and improving time extrapolation when dealing with latents and being data-free. They evaluate their method on two different datasets with model-based and data-driven baselines on different configurations and also show that PiDo is able to be transferred to downstream tasks.

**Strengths:**

The paper is well written, easy to follow and straightforward. The authors clearly identify what are the actual problems with data-driven and data-free approaches.

The approach explores an encode-process-decode strategy for learning dynamical systems like existing data-driven approaches, but propose to train it with a physic-informed loss, to remain data-free. It thus allows to be data-free while generalizing to different ICs and PDE coefficients unlike PINNs.

They propose two regularizations methods to improve time extrapolation and stabilizing PINNs training, closing the gap with data-driven approaches.

**Weaknesses:**

The paper is positioned at the intersection of EDMs, NOs and PINNs. However, the targeted task seems to be solving parametric PDEs with a data-free approach. The paper should be compared to meta-learning frameworks such as [1, 2].  Instead, the authors both compare it to a data-driven method (Dino) and model-based (PI-DeepONet, PINODE, MAD); each method seems instead to tackle a different task here.

For auto-decoding, if my understanding is correct, only one gradient descent step is needed in order to estimate new latents $c_0^i$. I am a bit concerned with the approximation made. How many steps do you actually need to find the optimal latent? How do you explain that such approximation still lead to very good perf? It could be nice to add some ablation studies comparing the impact of the number of gradient steps for learning parametric PDEs.

The visualisation in Figure 2 is not very well described. What do (green) training  and (red) test steps refer to here for a same trajectory? Each NS frame corresponds to a different time-step? It should be made clearer at least in the figure and its title to better justify the necessity of adding a regularization term.

Cho et al., Hypernetwork-based Meta-Learning for Low-Rank Physics-Informed Neural Networks, NeurIPS, 2023.

Belbute-Peres et al, HyperPINN: Learning parameterized differential equations with physics-informed hypernetworks, NeurIPS, 2021.

**Questions:**

PINNs approaches are known to face failure modes for certain PDEs. Are these failure modes also faced by PiDo or the use of a neural ode can help in that case?

Did you try to see if PiDo is able to generalize outside the training data distribution for new unseen PDE coefficients?

---

### Official Review · Reviewer_Ti5r · 2024-11-02

**Soundness:** 3
**Presentation:** 2
**Contribution:** 2
**Rating:** 3
**Confidence:** 4

**Summary:**

The author proposes a physics-informed dynamic representation learner that incorporates a temporal dynamic model to address the limited generalization ability of physics-informed partial differential equation solvers. Two regularization techniques are proposed to improve training stability and extrapolation ability, with significant experimental results.

**Strengths:**

1、The comparison of physics information neural partial differential equation solvers is very detailed, and the summary of previous literature is very comprehensive
2、The method proposed by the author utilizes an explicit dynamic model to give the partial differential equation solver significant advantages in extrapolation ability and training stability, solving optimization barriers related to physics information loss
3、From the current experimental results, the method proposed by the author has achieved significant improvement compared to existing methods

**Weaknesses:**

1、The number of experiments is not sufficient, and the selected equation types are also not sufficient.
2、Some spelling in the article needs to be carefully checked, such as in line 175

**Questions:**

Question
1、Further experiments are needed to support this model；
2、This article applies the Temporal dynamics model, which is not very attractive as the innovation point of the article

---

### Official Review · Reviewer_x2de · 2024-11-03

**Soundness:** 3
**Presentation:** 3
**Contribution:** 3
**Rating:** 6
**Confidence:** 3

**Summary:**

The paper introduces a new way of solving PDEs using neural networks. The main innovation of the Physics Informed Dynamics representatiOn learner (PIDO) presented in the paper is to encode the entire trajectory of the PDE (as opposed to only the initial state) in a latent space. This allows the dynamics of the system to be learned in this lower-dimensional latent space. The latent space view also helps design regularization terms that aid with training these networks. The authors show that PIDO produces more accurate solutions compared to competing methods and has better generalization properties.

**Strengths:**

**Originality**
* the method in the paper encourages generalization by learning a latent space that is useful for training the appropriate dynamics model across multiple choices of PDE parameters and time horizons.
* training is made more robust to extrapolation by combatting embedding drift. This is done by adding a loss term that encourages the latent-space embedding to be close to its counterpart when round-tripping through the decoder and encoder.

**Quality**
* the method proposed in the paper seems to significantly improve on existing methods in terms of accuracy of the solutions.

**Clarity**
* the paper is well-written and provides adequate background info to allow a non-expert to understand it.

**Significance**
* neural network PDE solvers are an important tool for addressing topics in various scientific fields (e.g., astrophysics) that require solving large numbers of equations as quickly as possible.
* the specific method in this paper can generalize across classes of PDEs that share the same functional form but have different coefficients, which can be useful for, e.g., solving inverse problems.

**Weaknesses:**

1. I think the paper could use a better explanation of how network and training parameters were chosen, especially when comparing different methods:
* Was any hyperparameter optimization done for any of the methods discussed in the paper? If not, it seems difficult to compare their performance.
* When comparing to data-driven approaches, my understanding is that those methods use a different training set (one containing actual full solutions of the PDE as opposed to physics-informed training used by PIDO). How do the authors then ensure a fair comparison between the two methods?
2. Figure 1 is hard to understand, I was only able to decipher it after reading the paper.
* I would move the "DYN" box over the unrolling arrow that goes from $c_0$ to the entire sequence $c_{0:N}$.
* I would split the unrolling arrow into two, one for coefficients $\alpha$ and one for $\alpha'$.
* The decoding of the solution curves should yield different results for $\alpha$ and for $\alpha'$.
* It's confusing that in the figure, for the initial conditions the decoder takes only the latent state $c_0$, while later it also takes a position $x$. Perhaps it can be indicated somehow that in the former case, we're evaluating the decoder at multiple positions.
3. Limitations are currently found in appendix D. Ideally these would be discussed in the paper itself.
4. Minor:
* in Fig. 5, the x-axis should be labeled.
* on line 175: missing "I": "[I]n this section"

**Questions:**

1. The errors from table 2 seem relatively high (a few are over 5%).  How much can we hope to lower this by, e.g., increasing the size of the network, training for longer, etc.? Do we reach a plateau, or is it in principle possible to achieve excellent accuracy given enough time and processing power?

---

> ### Comment · Area_Chair_5JzU · 2024-11-21
> **Comment of significance.**
>
> Dear Reviewer x2de,
>
> As I reading the reviews, it caught my attention that the one of the main innovation is : "This is done by adding a loss term that encourages the latent-space embedding to be close to its counterpart when round-tripping through the decoder and encoder."
>
> This seems quite close to the premise of [1] (which the authors cite in the manuscript). Is there any substantial difference between these approaches? Did the authors nuance their approach vis-à-vis the existing literature?
>
> Thank you again for your time and expertise.
>
> AC
>
> References:
>
> [1] Evolve smoothly, fit consistently: Learning smooth latent dynamics for advection-dominated systems. Wan et al. ICLR 2023

---

### Comment · Area_Chair_5JzU · 2024-11-26
**Rebuttal's Deadline Approaching**

Dear Authors,

As the author-reviewer discussion period is approaching its end, I strongly encourage you to read the reviews and engage with the reviewers to ensure the message of your paper has been appropriately conveyed and any outstanding questions have been resolved.

This is a crucial step, as it ensures that both reviewers and authors are on the same page regarding the paper's strengths and areas for improvement.

Thank you again for your submission.

Best regards,

AC

---

### Note · Authors · 2024-11-26

**Comment:**

Dear AC and reviewers,

After careful consideration, we have decided to withdraw the manuscript to further improve the quality of the work. We sincerely appreciate the valuable feedback and constructive suggestions provided by the reviewers, which have been instrumental in identifying areas for enhancement. We plan to incorporate these insights and conduct additional revisions to strengthen the study.

Thank you for your time and understanding.


Best regards,

Authors

**Withdrawal Confirmation:**

I have read and agree with the venue's withdrawal policy on behalf of myself and my co-authors.